# Generating Indicators of Disruptive Innovation Using Big Data

Roger C. Brackin [1],*, Michael J. Jackson [1], Andrew Leyshon [1], Jeremy G. Morley [2] and Sarah Jewitt [1]

1    School of Geography, University of Nottingham, Nottingham NG7 2RD, UK
2    Ordnance Survey, Southampton SO16 0AS, UK
*    Correspondence: roger.brackin@nottingham.ac.uk

**Abstract:** Technological evolution and its potential impacts are of significant interest to governments, corporate organizations and for academic enquiry; but assessments of technology progression are often highly subjective. This paper prototypes potential objective measures to assess technology progression using internet-based data. These measures may help reduce the subjective nature of such assessments and, in conjunction with other techniques, reduce the uncertainty of technology progression assessment. The paper examines one part of the technology ecosystem, namely, academic research and publications. It uses analytics performed against a large body of academic paper abstracts and metadata published over 20 years to propose and demonstrate candidate indicators of technology progression. Measures prototyped are: (i) overall occurrence of technologies used over time in research, (ii) the fields in which this use was made; (iii) the geographic spread of specific technologies within research and (iv) the clustering of technology research over time. An outcome of the analysis is an ability to assess the measures of technology progression against a set of inputs and a set of commentaries and forecasts made publicly in the subject area over the last 20 years. The potential automated indicators of research are discussed together with other indicators which might help working groups in assessing technology progression using more quantitative methods.

**Keywords:** disruptive; innovation; technology; assessment; big data; unified technology progression modelling

## 1. Introduction

Many groups, including governmental, academic, and commercial are interested in identifying the direction of technological evolution. The goals can be, respectively:

- To allow governments to focus support on technologies most likely to be significant, while also defending against potential evolving technology threats.
- To provide academic institutions or other organizations planning research and development (R&D) with a measure against which to assess research proposals or research and development plans.
- To help commercial organizations focus investment and product development in areas which are likely to progress and integrate with evolving technology development.

For these reasons many organizations undertake technology assessment activities both as a routine process and on an ad hoc basis. The goals are typically to assess how mature a technology is and how fast it is developing as well as to assess likely future progression. This paper is focused on the first two of these goals. However, understanding the current state of a technology's development can help groups also interested in the likely future progression.

A common approach to assessing technology progression is to assemble a group of experts and discuss which new technologies are likely to make a significant impact on or to be of benefit to the target community. It is then common to further analyze this manually and collate a report. The authors of this paper have organized, chaired, and participated in many such seminars and provided reports on technology progression to

key government customers for over 20 years. In these activities it has been observed that there is a lack of effective tools or techniques to support this activity. There are many consultancy organizations and pundits who undertake these activities for corporate and government clients, but their methods are only published to a limited degree, e.g., Linden & Fenn (2003) [1]. The US National Research Council undertook the most extensive assessment of the field (NRC, 2009 [2]; NRC, 2010 [3]) but there has been little published work since in providing an integrated methodology for technology assessment. There are many authors who, in a social science/business context, describe aspects of technology progression (Bower & Christensen (1995) [4], Arthur (2009) [5], Mazzucato (2011) [6], and many others. However, they are currently proven only by limited examples. If they are valid theories, their application to assist in technology assessment would offer significant value. Lastly, there are big data, machine learning approaches to assess trends. There is a body of research which is addressing trends, although much of it focusses on the later phases of technology development such as financial progress, for example Gerasimos (2017) [7] and Parker (2010) [8]. This paper is focused on indicators of the earlier phase of technology development within research and offers proof of concept indicators which may help support technology progression assessment.

After briefly reviewing published work in this field, typically splitting along the lines of theoretical models, current manual approaches and big data/analytics approaches (Section 2) this paper suggests a research question and the approach to addressing this question (Section 3). The results are then described (Section 4). Finally, the implications of this work and proposed future work are covered in Section 5.

## 2. Modelling Technology Progression

This section considers three main areas of related research in the field of technology progression assessment. These are:

- Current approaches to technology progression assessment, such as NRC (2009 [2], 2010 [3]) There is a dearth of formal work in this field and little best practice documented, hence a desire by the authors to improve on this by providing tools.
- The concepts of technology progression and disruptive technology, which were formed by authors such as Christensen (1997) [9] and Arthur (2009) [5], and enhanced most recently by authors such as Langley & Leyshon (2017) [10].
- The application internet/big data approaches to assessment of technology progression often in specific areas, e.g., Treiblmaier (2021) [11] in 'Future Internet'. There is much related work but little that has a strong link to the goal of supporting an integrated assessment capability linking to the above two topic areas.

Whilst this paper is largely focused on techniques in the third area area listed above (automated approaches) a key goal is to do this in the context of supporting an improvement of the first item (improving current approaches) and exploiting the second (technology progression theories). The following subsections examine these three aspects of existing research in more detail.

### 2.1. Current Approaches to Technology Progression Assessment

The first approach is the use of panels of subject area experts on an ad hoc basis to suggest technologies or technology trends which could have a significant influence on society or commerce. Activity can range from short one day or two-day seminars to more significant consultancy exercises taking months of effort. They can also entail questionnaires and interviews. This paper's authors have been involved in initiatives at both extremes for agencies such as the European Space Agency, the UK Defence Science and Technology Laboratories and the Open Geospatial Consortium.

Organizations such as Gartner and Deloitte, and publications such as the Economist, Forbes and the New Scientist attempt a longer-term assessment of technology progression. Gartner, on their website, suggests they use multiple techniques, although methodological details are limited. However, they have both public and more detailed private assessments.

The Economist examines technology on a regular basis in the weekly newspaper with quarterly and occasional special sections.

The National Research Council (NRC) undertook a significant review of technology progression and to some degree forecasting approaches over two years (2009–2010). The reports NRC (2009) [2] and NRC (2010) [3] suggest a persistent monitoring system exploiting multiple approaches to assessment. These include automated and human based initiatives. Hang & Chen (2010) [12] describes an assessment framework to allow more quantified and repeatable judgements around disruptive innovation. Radosevic (2016) [13] discusses the theory and metrics of technology upgrading and presents some interesting aspects of the measurement of technology progression.

Different groups, government, academia and business have different requirements ranging from the very broad, so called horizon scanning approaches to very specific in-sector analyses or even a focus on a single or limited range of products. In some cases, there are also different goals. Amanatidou (2012) [14] describes two different requirements and therefore approaches as exploratory or issue-centred scanning, respectively and the two are quite different (the former with much higher uncertainty). Jovic (2020) [15] provides an example of a domain specific investigation into disruptive innovation, in transport management systems.

### 2.2. Theories of Technology Progression

In assessing 'technology' we need to discuss the scope of this term. Technology covers a scope from specific (e.g., a sports watch) to broad, e.g., the 'internet connected home'. Arthur (2009) [5] considers many of the nuances of technology as a concept, but in principle it can be a small isolated application of science or a broad, far-reaching concept. Another common term used in relation to technology is innovation. The two terms technology and innovation-are used relatively interchangeably with innovation being used rather more generally (for example by ranking countries on their level of overall innovation (WIPO, 2021). Originally created by Bower & Christensen (1995) [4], the term disruptive technology is often also referred to as disruptive innovation.

Various writers have considered parts of the technology ecosystem including Christensen (1997) [9] who develops the concept of disruptive technologies, Arthur (2009) [5] who outlines the ecosystem of producers and consumers, and Mazzucato (2011) [6] who discusses the effect of government research on supplying technological advancements. Arthur (2009) [5] attempts to frame the technology ecosystem, see Figure 1; this identifies the elements of a technology and the organizational players and interactions which occur, described at a macro level.

An important aspect of modern technologies is their connectivity and therefore the level of leverage they achieve (Langley & Leyshon 2017 [10]). The concept of a platform is a derivative of the general case of a composition of technology described above and one that is particularly relevant to information technology development (Simon, 2011 [17]; Srnicek, 2017 [18]). The platform is a collection of base technologies which allow other technologies to grow at an accelerated rate.

Smartphones support the display of maps, but the maps are served from a central server to a massive number of users; similarly with speech recognition is performed on central infrastructure rather than on the device. In formal terms then, we potentially should separate 'smartphone' from 'smartphone platform' (Xuetao, 2013) [19]. In general, we do not talk about the 'smartphone platform'. Brackin et al. (2019) [16] discusses this issue in more detail. Taking the example of the smartphone from Brackin et al. (2019) [16], a whole range of technologies was necessary to allow it to work effectively (see Figure 2). The value of smartphone devices comes from the massive infrastructure in which they exist (Xuetao, 2013) [19].

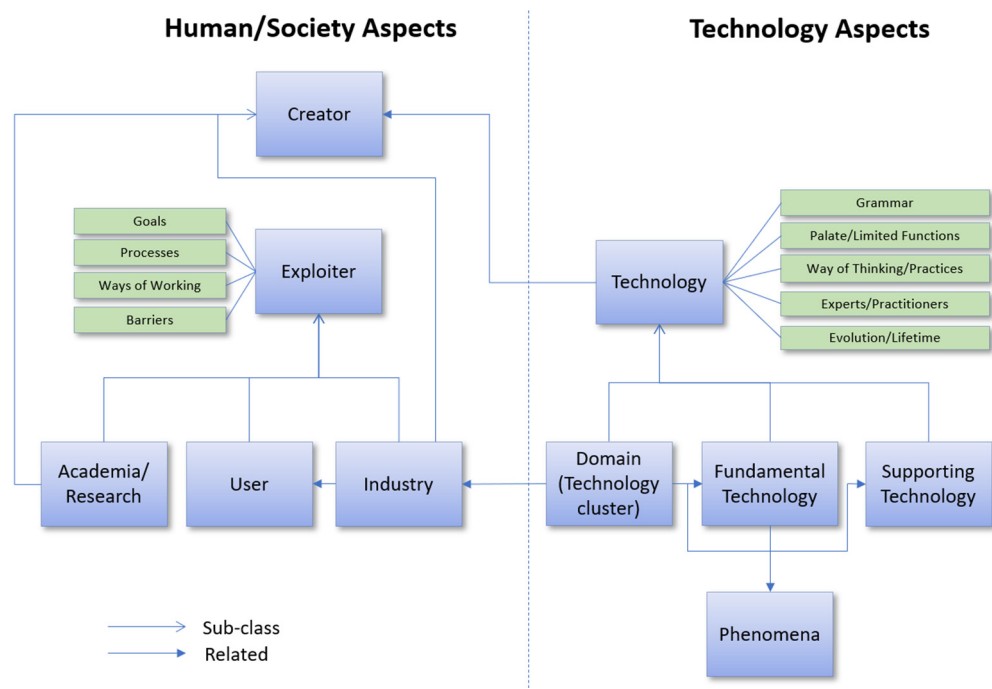

**Figure 1.** Arthur's concept of the technology eco-system, Brackin et al. (2019) [16].

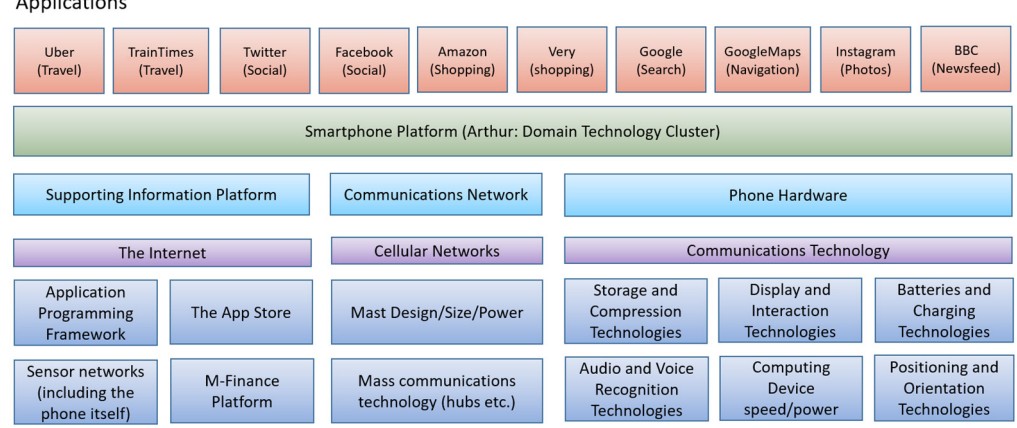

**Figure 2.** Technologies supporting the smartphone platform.

Brackin et al. (2019) [16] postulates that this form of technology development, clustering to form a new composite technology, may be easier to identify as it is evolutionary rather than a completely new technology. Identifying a grouping of technologies early could help in identifying technology trends and likely disruption. The smartphone grew from simply providing a communications device to a capability supporting a range of other applications relatively quickly. Identifying this 'move' in applicability could also help identify technology direction and evolution. Thus, a key focus of this paper is how we can potentially measure such effects as soon as possible to aid in decision making.

In assessing technology progression there is also a need to identify technology synonyms, or changes. For example, unmanned aerial vehicle, UAV and drone are used interchangeably. However, understanding whether a name change indicates a functional change is also an issue.

The evolution from science to technology is potentially of interest, and the barrier is, as with everything at a macro level, relatively blurred. Science institutions have to some extent moved away from purely fundamental research and are now more focused on delivering results and receiving significantly more funding from industry. Industry is also investing in primary research (because of the significant capital value that exists in organizations

such as Amazon, Google and Apple). Mazzucato (2011) [6] suggests that many commercial products (such as the Apple iPhone), only exist because of numerous government research outputs which were then exploited by Apple (see Brackin et al. (2019) [16]). The implication of technology (the precursor to products) being created in commercial organizations is that it is less visible. The experimentation described in this paper highlights issues around this.

The concept of the platform provides a mechanism for allowing accelerated creation of technology and comes in various forms. Langley & Leyshon (2017) [10] describes several types of platforms all of which have a technology component. These include technological (e.g., the smartphone with its deployment environment), payment systems and financial models such as a crowdfunding platform which accelerates the rate funding can be obtained. Srnicek (2017) [18] comments on the challenges of the platform, and MOK & Brynjolsson (2017) [20] consider the business implications of the platform. The topic is discussed further in Simon (2011) [17]. The platform, once established, allows multiple other technologies to be deployed very quickly. This topic is dealt with further in Brackin et al. (2019) [16].

Mokyr (2016) [21] describes the element of randomness that causes technology creation, a process he describes as 'tinkering'; this is almost Darwinian in its nature. Arthur (1994) [22] undertakes a detailed treatment of the potential randomness of initial technology progression, which he suggests slowly becomes more predictable as a particular alternative progresses ahead of another.

Masters & Thiel (2014) [23] in 'Zero to One' suggests that having established a technology in a limited area it can then be transitioned to a broader market with disruptive effect. This is consistent with Christensen (1997) [9] who offers the proposition that highly disruptive technologies sometimes come from a technology which 'slides' in from one specific domain to a more general one. An example of this is the iPod which established a niche and then grew to become the iPhone, causing massive disruption in the mobile phone market, Mallinson (2015) [24].

Another issue, particularly significant to governments who want to capture the value of technology is where (geographically) it is created, e.g., HM Government Strategy (2017) [25]. Important aspects are where a technology is created and then how it is manufactured and ultimately used. This is an interplay of innovation, the requirements, and the production/distribution costs of a technology/product.

*2.3. Assessing Technology Progression*

The two studies conducted by the US National Research Council (NRC, 2009 [2]; NRC, 2010 [3]) did consider potential integrated and persistent approaches to technology assessment based on a mixture of automated and human analysis; this work never progressed. (The research lead and editor of NRC (2009) [2] and NRC (2010) [3] was contacted and confirmed that the work terminated in 2010.)

Brackin et al. (2019) [16] suggests an outline for a unified model of a technology ecosystem, integrating the various approaches of the authors mentioned in Section 2.2. Arthur (2009) [5] offers the most complete macro model of technology anatomy, which is mapped out graphically in Figure 1. Brackin et al. (2019) [16] also suggested that measurement systems or indicators are necessary to validate, calibrate and assess the state of various elements of models of the technology ecosystem. These indicators could be like the source used in weather and climate modelling (Parker 2010) [8] or in financial systems (Gerasimos 2017) [7]. Both modelling domains involve massive numbers of complex input variables and measurements, complex predictive models, and the need to particularly address calibrating for the level of uncertainty. The alignment of the model forecast with the measured state is continually assessed and the model output continually validated and re-calibrated. A key element of the progression of technology is the degree to which component technologies come together to solve a range of problems forming in effect a hybrid technology. This form of technology development is also potentially measurable.

Brackin et al. (2019) [16] considered the rate of technology growth overall and in relation to other technologies, the geographic origin and spread of technology, the interlinking

to form new technologies (using the smartphone as an example as it is a hybrid of multiple technologies) and the application of a technology to a new field. Each of these topics can be found in the works of the authors noted above, and in some way measuring them would potentially provide useful indicators.

Until recently there has been a significant dearth of approaches to automating the process of identifying new technologies of interest. While both NRC reports (NRC, 2009 [2]; NRC, 2010 [3]) discuss it in general terms as does Tanaka (2005) [26] there does not seem to be a significant body of literature attempting it. One issue is that to identify technologies 'appearing over the horizon' via automated means is likely to require vast complexity, akin to language translation. Technologies in this stage also fit in the Arthur (2009) [5] model of path dependence in the high uncertainty stage.

Specific areas are assessed by Treiblmaier (2022) [11] which considers automated assessment of technology progression using Bitcoin as the specific example of potential disruption. Claus (2022) [27] applies natural language processing and cognitive networks to investor day transcripts to assess progression in Insurance.

Martin & Moodysson (2011) [28] and Asheim (2009) [29] also consider the issue of technology progression. Nathan (2014) [30] looks at London's 'Tech City' development and Schmidt (2015) [31] considers spatial localization in relation to knowledge generation. The 2020 World Intellectual Property Organization (WIPO) report ranks countries by level of innovation, as well as to some degree sector biases, i.e., specific types of innovation of interest to specific countries or localities.

More general approaches have also been developed recently. Calleja-Sanz (2021) [32] provides a review of some of these. Dellermann (2021) [33] proposes a machine learning approach using approaches that might be expected, Logistic Regression, Naives Bayes, Neural networks and ensemble analysis commonly used in meteorological models to assess sensitivity of results to input parameters. The suggestion is also that the approach should use both machine and collective intelligence. At present the proposed method has not been tested, so that seems to be the next step. Calleja-Sanz (2020) [32] also appears to address this area although this is more of a review of manual and computer techniques rather than new research. Carbonell (2018) [34] looks at patterns of technology progression, as does Claus (2022) [27]. Dernis (2016) [35] looks at co-development trajectories of technology which aligns with many of the concepts of Arthur (2009) [5] related to hybrid technology development and one of the key measures in this paper. Lastly, Chang (2022) [36] looks at technology progression by exploiting patent mapping and topic trend analysis.

### 2.4. Summary of Existing Research

The above research represents the three related aspects of this field (shown in Figure 3). The practical goal and best practice in assessing technology progression, the theories of technology structure and progression largely developed in a social science/business arena, and the approaches which build on analytical techniques.

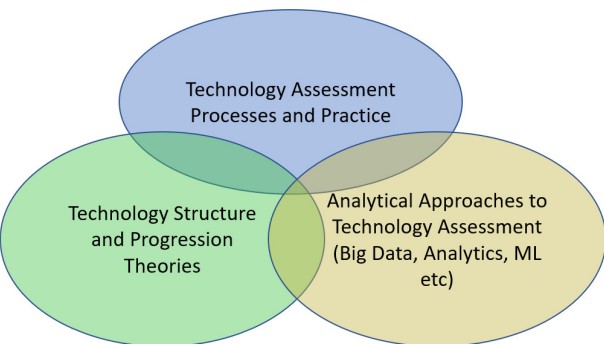

**Figure 3.** Three areas of Technology Progression Research.

This paper seeks to offer analytical views and measures in the third category in Figure 3 (Analytical Approaches), but which try to draw on the theoretical work in business and social science (Technology Structure and Progression) and help support the process of Technology Assessment. The opportunity is to potentially integrate the measures related to research progression, with other measures described in the references above, including patents and stock market performance, to provide a rounded picture of technology progression. This broader goal is considered in Section 5.

*2.5. Structure and Goal of This Paper*

This paper addresses the primary research question, in the context of historic analysis of available data:

**"To what degree is it possible to identify objective indicators of technology progression based on historic data from academic research".**

The paper also examines whether these measures/indicators align with other subjective assessments over time and with the many theories of technology evolution. The outputs of analyses are in some cases presented graphically or in tabular form, allowing the opportunity to assess them, but also specific numerical assessments are calculated.

Emphasis is placed on whether it is possible to use the level of research being undertaken on a technology or technologies over time (available in open data) to provide indicators of technology progression by comparing the calculation that could have been made over two decades with the reality identified by other sources over that time. At this stage the goal is not a fully formed formal method, but an assessment of the potential viability of generating objective analytics/measures.

Overall, if a set of indicators could support even a small narrowing of the vast range of potential technology outcomes or help validate other assessments, then it is suggested they would offer significant value.

Section 3, describes the method and approach to providing an indicator as well as the key input information available from open sources. This section then sets out the various processes of harvesting information from the internet and analyzing it. The methods section also describes and shows the various visualizations created to allow the analysis to be explored by a user interested in the assessment. This section also outlines how the analysis system was validated.

Section 4, describes the specific experiments undertaken, which were to assess the indicator against two sources of technology forecasts over the last two decades (Gartner and the Economist). It also considers the success of the indicators in providing a resource to potential users.

The final section of the paper draws together conclusions from the development of the indicators and suggests further work that should be undertaken.

**3. Method/Approach**

To test the research question, indicators of the progression of a set of technologies within research were developed. These indicators measured the prevalence and relationship of a set of 'input technologies' in academic material, using open and public data, to produce measures of the progression. The input was a set of technology terms, with a date against each showing when they were first identified. Several sources of such terms were used, as described below. The sources chosen offered technology terms over a period of years (between 10 and 20), allowing the opportunity for historic analysis.

This paper seeks not to devise methods of identifying new technologies themselves, but instead to assess candidate technologies over time. The measures of technology progression devised were:

- The level of activity or interest in a technology topic over time in academic papers.
- The geographical progression of technology topics over time, where they begin and how they progress geographically.

- The correlation between the progression of the technology topics, identifying relationships between them.
- The occurrence of technologies over time in research related to specific subject areas such as medicine, social sciences, or law.

The approach taken is to exploit available information on technology progression over two decades (i.e., perform retrospective analysis).

Given the successful technologies from the last two decades are now known, the method's success in identifying the growth of a set of technologies over others can be assessed over the period proposed (two decades from 2000 to 2020). The key elements of the experimentation were:

- A list of technologies which have evolved in the last 20 years, and where available an indication of when they became mature.
- A representative set of information describing the academic research undertaken in the last 20 years. The abstract of a paper is sufficient to understand its subject/aim, it was decided that reviewing the title/abstract and publication date would be sufficient.
- Software to ingest and process the reference sources of technology progression and to provide relevant analytics to allow several indicators to be produced and assessed.
- A capability to compare the indicator output for each year over the last 20 years to the external technology sources chosen (in 1 above) and thus assess the viability of the approach.

### 3.1. Selection of Technology Sources

Sources of technology progression which were considered as potential inputs were measured against the following criteria:

- Publicly available commentary on technology
- A widely recognized source of technology assessment
- Available for at least 10 years of the last two decades.

Two enduring sources meeting the criteria above were business research organization Gartner and the Economist magazine. Both Gartner and the Economist have published information on technology progression over the period considered. Gartner has openly published (annually) a technology assessment over the last 20 years (the annotated hype cycle diagram), which is covered further in the third research question. The Economist has a regular section covering science and technology, and a regular quarterly bulletin.

With both Gartner and the Economist, a progression over time of the technologies they highlight-and in Gartner's case the relative maturity were available.

Gartner's public offering, the 'hype cycle', published annually, is a list of technologies identified on a curve showing their assessed point of evolution (Figure 4).

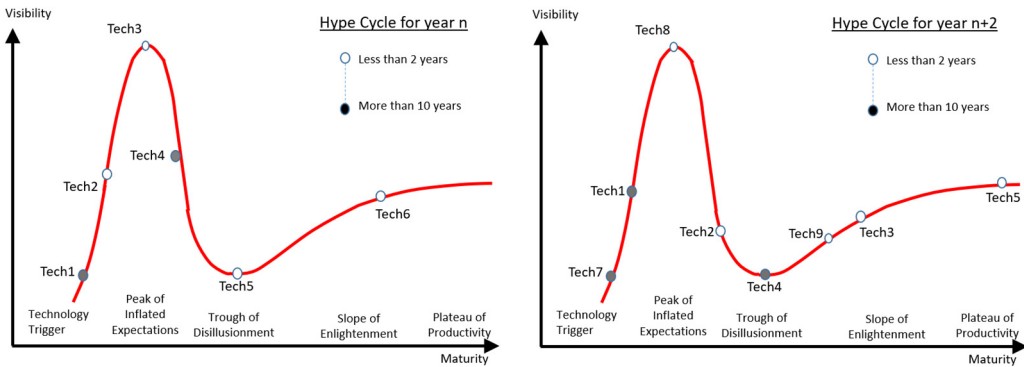

**Figure 4.** Gartner 'hype cycle' model.

The concept of the 'hype cycle' is outlined in Linden & Fenn (2003) [1]. Technology topics low on the cycle (to the left) are considered immature candidates and topics nearer

the right of the diagram are considered a success. Using the technology topics identified each year (either the first occurrence or the mature end of the graph) gives a reference for comparison (when the technology was noted by Gartner's assessment and when they believe it became 'mature').

Each week the Economist newspaper looks at the implications of various technologies. It also includes regular quarterly reviews which focus on emerging technology topics. Copies of the Economist were available for two decades.

By reviewing Gartner's hype cycles over the two decades chosen and the Economist newspaper over the same period a list of technology terms was identified together with the date occurrence. This was a relatively manual process, and it is difficult to ensure total reliability in detection of terms. It should therefore be noted that these sources were simply used to generate representative lists of technology for each year. This paper does not try to assess the accuracy of these sources. They potentially offer between them an indication of what technologies were considered as emerging from 2000 to 2020.

### 3.2. Selection of Academic Paper Data

Paper metadata (paper title, abstract and publication date) was used to perform the analysis. If a technology term was not identified in the title or abstract, then it was unlikely to be a significant part of the research. Other information—such as the authors, the geographic location of the research and the topic area of the paper (e.g., medicine) would also be beneficial. Paper metadata is generally publicly available whereas the full paper is often not. Thus, a wider coverage could be achieved simply using the metadata.

Several sources of academic paper summaries were considered including Google Scholar and academic reference organizations such as Elsevier. For this work though, it was important to be able to harvest information programmatically, which is problematic with most sources (e.g., Google) as automated processing requires special permissions; this was found to be true of most commercial sources. However, most academic organizations are now contributing to the Open Archive Initiative (OAI), which provides a common, machine access mechanism definition with which organizations comply, and a list of contributing organizations and the URLs for their OAI access point. The OAI started in the late 1990s and is now supported by a significant number of academic organizations and journals globally. There are currently around 7000 stated member organizations. Some of these do not currently publish an OAI URL or have some form of protection on their URL. The latter may be resolvable with a specific request to the organization but without such action, 3219 sites can be accessed on an open basis (this number was established by interrogation and validation of each URL response automatically) and these have all been used in the experiment. Although many academic and not for profit organizations have joined the OAI initiative later than the 1990s, they are typically loading all historic papers. This means that at least two decades of paper metadata is typically available for each new organization as it comes online.

### 3.3. Analysis Framework

The analysis of the paper abstracts required the development of several pieces of technology. Components were developed as needed to undertake the necessary processes. These are shown in Figure 5.

The framework was designed to allow further extension with additional data sources, harvesters and analyses. Further papers will describe these analyses and their results as the research progresses.

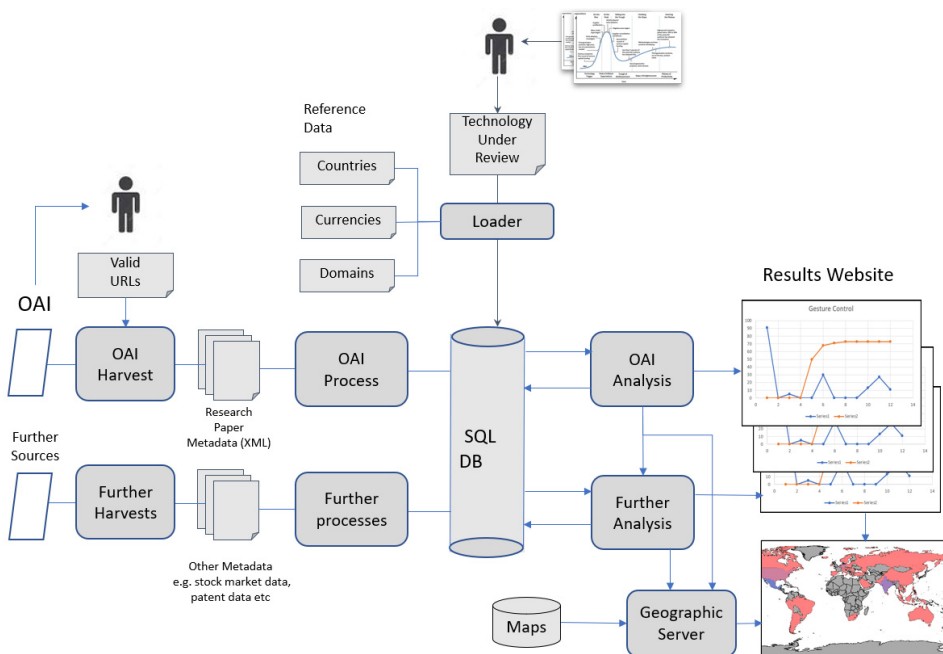

**Figure 5.** The experiment Analytic Framework.

The technology choices were made specifically for the type of data and measures an analysis required. For example, because the research paper data were largely tabular, and efficient text searching was needed, a relational storage model and thus a RDBMS was used. Some of the outputs are spatial and others network related. The framework is extensible to integrate other forms of analysis—for example, it currently generates HTML and csv but could easily output a graph representation of the relationship data (in for example GraphQL or RDF).

### 3.4. Harvester/Loader

After considerable investigation and experimentation, the OAI resources were determined to be the most useful and complete open and exploitable source of paper abstracts, and it proved to be an extensive resource for the experiment. The amount of data (with 40,000,000 papers' metadata, from 3219 OAI libraries), was of the scale needed.

The harvesting and population of a database was initially time consuming (even though largely automated), but the concept of 'incremental refresh' means the resource is easy to keep up to date. The approach of downloading data rather than using it online meant that various experiments could be performed with no concerns about being blocked by websites for excessive requests for data. It also made repeatable experiments possible.

Because of the need for repeated and detailed analysis and the amount of data, it was necessary to harvest the paper metadata into a central database (this avoided overloading the individual organizations' websites). A harvester program was implemented to retrieve the records. This connected to each organization, requested the metadata for each available paper, and stored this in a relational database. The first harvest was performed in early 2019 and in early 2021 papers added since the 2019 harvest (including from new contributing organizations) were added. In total 41,974,693 paper abstracts were harvested from the 3219 organizations' OAI interfaces, with publication dates from 1994, and with origins in over 100 countries. The initial harvesting and loading process (in 2019) took approximately three weeks and the update process (in 2021), approximately 5 days.

### 3.5. Data Cleansing/Enhancement

With the number of organizations included, and the amount of data published and subsequently harvested, significant data gaps/issues exist; thus, a cleaning process was performed prior to analysis. For example, a missing and badly formatted publication date

could be problematic. For the analysis, only the publication year was required. If the publication date was not valid, as a backup the year of entry into the OAI library was used as an alternative.

Geographic location is not present in OAI Paper metadata. An inference can be made by looking at the publishing organization. The location for this had to be derived from the publishers URL (for example ac.uk is unique to the UK) or if still unresolved by manual examination (e.g., the description, e.g., 'Stanford' identifies a paper as US). Approximately 60% of the 3219 organizations country could be inferred from the URL, the remainder were classified manually or left as 'worldwide'.

There is the risk of significant duplication as papers are often published both in a journal and concurrently on a university website. Since the title is typically unchanged when republished, removal of duplicates is relatively easy. Queries were developed that could mark them as excluded from the analytics search.

In addition to the academic papers, the manually created spreadsheets of technology terms and occurrence dates from the reference sets were cleaned and consistently formatted for input into the analysis process.

### 3.6. Database Design

The database objects used to support the initial experiment are shown in Figure 6.

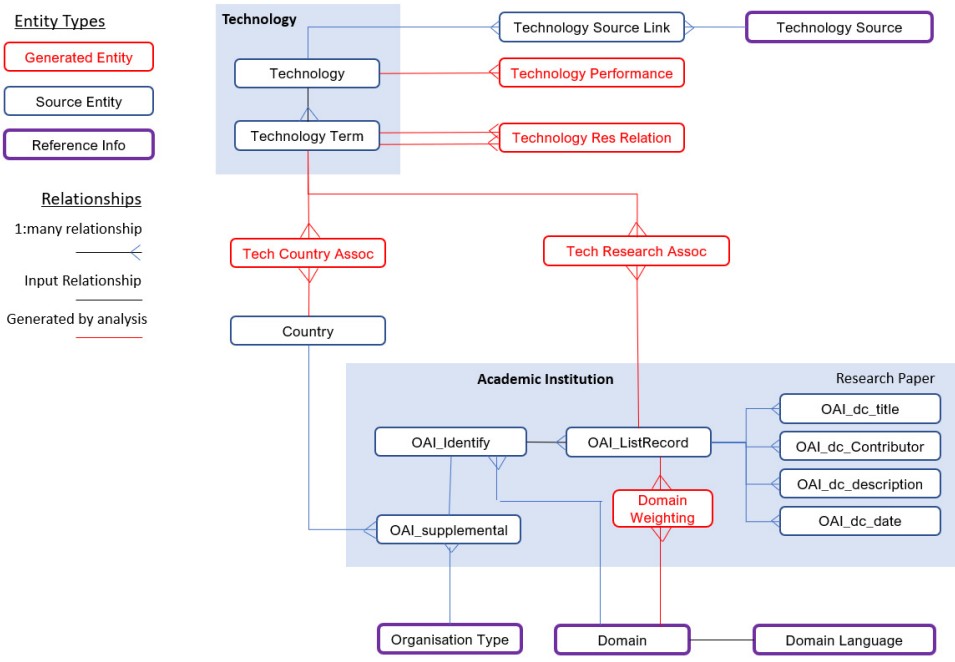

**Figure 6.** Experiment Database Design.

The elements of the model include the following elements, populated with data either by the harvest process or by the analytic process (see the key above):

- Source entities are items which are loaded from two sources, research data from the Open Archive Initiative (OAI) and technology terms from internet sources and journal review.
- Reference information such as domains/sectors and technology classification types and country codes are loaded from reference files.
- Generated Entities include computed data and metrics, including associations between technologies and research.

### 3.7. Analysis and Results

For each technology term in the reference list (Gartner, Economist, etc.), a query was performed to obtain a list of papers in which the term appears in the abstract. The input to the process was a file of the form shown in Table 1.

**Table 1.** Technology Topic Input.

| Technology Term | First Occurrence | Source | Synonym(1) | Synonym(n) |
|---|---|---|---|---|
| Cloud Computing | 2008 | Source1 | | |
| Virtual Reality | 2011 | Source1 | VR | |
| Service Oriented Architecture | 2016 | Source1 | SOA | |
| Unmanned Aerial Vehicle | 2016 | Source1 | UAV | Drone |
| Organic Light Emitting Diode | 2012 | Source1 | OLED | |
| 3D Printing | 2011 | Source1 | 3d Printer | |

Queries were derived which analyzed both the formatted fields in the data (dates, locations, etc.), and scanned the free text in the title and abstract in the metadata for all papers. Several techniques were used to avoid falsely detecting short words in other words (e.g., 'VR' in manoeuVRe) by carefully conditioning the query and post processing.

The following measures were generated:

- Profile over time of a technology term (number of occurrences each year).
- Profile over time of the term within different subject areas (technology, medicine, law, etc.).
- Profile over time of the publication country of papers containing the term.
- Co-occurrence with other terms (how often do other technology terms appear in papers relating to one technology).

Each measure offers a different insight into technology progression—addressing the overall growth of a technology, how it starts to pervade different subject areas (applications), its geographic spread, and its alignment with other technologies. The latter supports identification of co-dependence, or shared relevance to a problem.

The results of the analysis are presented in textual and graphical form in a linked HTML structure (results website), allowing for the revision of the analysis of a given collection of technologies. In addition to the measures pages per technology, there is an overview page, providing an index for the results. A typical overview page output for a processing run, with a list of input technologies, is shown in Figure 7.

In addition to the human readable outputs (HTML) a series of data outputs were generated, comma separated variable (CSV), JavaScript object notation (JSON) and resource description framework (RDF) files per technology. These allow other tools to be used to assess the data.

To validate the results or review specific technology topic results, both the titles and the paper abstracts were presented in HTML for a given technology topic (Figure 8).

The abstract table is helpful in identifying erroneous detections, particularly with the implementation of highlighting showing where the detections occurred. A typical issue was, for example, the term 'OLED' (organic light emitting diode) is a common string in many words (e.g., pooled). This was easily avoided once identified (by conditioning the queries) to add spaces around small terms "oled" and "(oled)" and also some further processing. This is not perfect as it then misses some potential results but is more stable.

For a given set of reference technology topics (over 300 in total), all the analyses described in the previous section were calculated automatically and hyperlinked to the relevant technology topic presented in an HTML table as shown in Figure 7.

Summary of Results

| Technology Topic | Tech Topic Date | Term First Occurs | Predicted | Total Occurances | Occurances v Time | Term Countries | Geographic Distribution | Result Details | Related Technologies | Domains Using |
|---|---|---|---|---|---|---|---|---|---|---|
| 3d printing | 2015 | 2008 | Precursor (7 years) | 6407 | Graph | Table/graph | Year/Occurance | Titles/Abstracts | Table/Graph | Table/Graph |
| 5g wireless | 2016 | 2016 | Same Time | 243 | Graph | Table/graph | Year/Occurance | Titles/Abstracts | Table/Graph | Table/Graph |
| additive manufacturing | 2017 | 2000 | Precursor (17 years) | 5922 | Graph | Table/graph | Year/Occurance | Titles/Abstracts | Table/Graph | Table/Graph |
| ai in medicine | 2015 | 2011 | Precursor (4 years) | 14 | Graph | Table/graph | Year/Occurance | Titles/Abstracts | Table/Graph | Table/Graph |
| anti-malaria drugs | 2016 | | Not Found | 0 | --- | ---/--- | ---/--- | ---/--- | ---/ --- | --- /--- |
| artificial intelligence | 2015 | 1990 | Precursor (25 years) | 14447 | Graph | Table/graph | Year/Occurance | Titles/Abstracts | Table/Graph | Table/Graph |
| artificial neurons | 2016 | 2009 | Precursor (7 years) | 262 | Graph | Table/graph | Year/Occurance | Titles/Abstracts | Table/Graph | Table/Graph |
| augmented reality | 2016 | 2000 | Precursor (16 years) | 5425 | Graph | Table/graph | Year/Occurance | Titles/Abstracts | Table/Graph | Table/Graph |
| automated hypothesis generation | 2014 | 2021 | Not Precursor | 1 | Graph | Table/graph | Year/Occurance | Titles/Abstracts | Table/Graph | Table/Graph |
| autonomous vehicles | 2018 | 2007 | Precursor (11 years) | 2421 | Graph | Table/graph | Year/Occurance | Titles/Abstracts | Table/Graph | Table/Graph |
| battlefield nuclear energy | 2020 | | Not Found | 0 | --- | ---/--- | ---/--- | ---/--- | ---/ --- | --- /--- |
| behavioral biometrics | 2019 | 2006 | Precursor (13 years) | 58 | Graph | Table/graph | Year/Occurance | Titles/Abstracts | Table/Graph | Table/Graph |
| biotechnology | 2017 | 1990 | Precursor (27 years) | 21516 | Graph | Table/graph | Year/Occurance | Titles/Abstracts | Table/Graph | Table/Graph |
| bitcoin | 2017 | 2014 | Precursor (3 years) | 1712 | Graph | Table/graph | Year/Occurance | Titles/Abstracts | Table/Graph | Table/Graph |
| boring technology | 2017 | 2016 | Precursor (1 years) | 6 | Graph | Table/graph | Year/Occurance | Titles/Abstracts | Table/Graph | Table/Graph |
| brain scan | 2017 | 2003 | Precursor (14 years) | 457 | Graph | Table/graph | Year/Occurance | Titles/Abstracts | Table/Graph | Table/Graph |

**Figure 7.** Analysis summary and links page (HTML).

| L2756-F532-R78-D2021-02-06 | | 2019 | United States | Simulating secondary organic aerosol in a regional air quality model using the statistical oxidation model – Part 1: Assessing the influence of constrained multi-generational ageing | [x] Multi-generational oxidation of volatile organic compound (VOC) oxidation prod (SOA) compared to calculations that consider only the first few generations of oxida models that account for multi-generational oxidation (1) consider only functionalizat and (3) are added on top of existing parameterizations. The incomplete description o calculations for SOA. In this work, we used the statistical oxidation model (SOM) o: implications of multi-generational oxidation considering both functionalization and : California Institute of Technology (UCD/CIT) air quality model and applied to air qu using SOM were compared to SOA predictions generated by a traditional two-produ oxidation. Results show that SOA mass concentrations predicted by the UCD/CIT-S( are derived from the same chamber data. Since the two-product model does not expl parameterize the models captures the majority of the SOA mass formation from mul perturbs SOA concentrations by a factor of two and are probably a much stronger de SOM and two-product models, the SOM model predicts increased SOA contribution and monoterpene relative to the two-product model calculations. The SOA predicted qualitative agreement with volatility measurements of ambient OA. On account of it inclusion of oligomerization reactions, whereas the two-product model relies heavily scheme to model multi-generational oxidation within the framework of a two-produ considered. This hybrid scheme formed at least 3 times more SOA than the SOM du: material that strongly partitions to the particle phase. This finding suggests that these |
| L2008-F232-R25-D2021-02-06 | | 2021 | United Kingdom | The AeroCom evaluation and intercomparison of organic aerosol in global models | [x] This paper evaluates the current status of global modeling of the organic aerosol ( observations. Thirty-one global chemistry transport models (CTMs) and general circ The simulation of OA varies greatly between models in terms of the magnitude of pr of OA parameterizations (gas-particle partitioning, chemical aging, multiphase chem global OA simulation results has increased since earlier AeroCom experiments, mair highly uncertain, OA sources. Diversity of over one order of magnitude exists in the although the OA / OC ratio depends on OA sources and atmospheric processing, and models. The median global primary OA (POA) source strength is 56 Tg a−1 (range : Tg a−1). Among the models that take into account the semi-volatile SOA nature, the the models that calculate SOA in a more simplistic way (19 Tg a−1; range 13–20 Tg of 0.6–2.0 Tg and 4 between 2.0 and 3.8 Tg), with a median OA lifetime of 5.4 days OA/sulfate burden ratio is calculated to be 0.77; 13 models calculate a ratio lower th is 70 Tg a−1 (range 28–209 Tg a−1), which is on average 85% of the total OA depos individual field campaigns have been used for model evaluation. At urban locations, strength and seasonality. The combined model–measurements analysis suggests the ( |

**Figure 8.** Paper abstract output sample for the SOA topic.

Various steps were taken to validate the harvesting and analysis.

- An extensive logging of errors was implemented and retained, allowing all failures in processing to be reviewed.
- To test if the set of paper abstracts was representative, the papers present for each of the authors of this paper were searched for (as we are each aware of the papers we have published and so can check their presence, relevance, and publication dates). The results were as expected.

- The abstracts of a sample of occurrences of a term were examined to verify they were in general correct and not false detections (the highlighting of terms in the abstract helped here)
- The categorization of papers based on words in the abstract (as law, medicine, etc.) was verified by passing through papers from institutions specializing solely in one of the disciplines and checking that discipline scored highly.
- Association level was checked using two sets of terms which were generally unrelated. The expectation was to find close grouping within each set and little cross linking, which was the case.

While not exhaustive, these tests provide the basis for confidence in the results.

*3.8. Technology Used for the Experiment*

The experimental software was largely developed in the Java programming language. This included all key components (the harvester/database loader, the cleansing software and the analysis component). Supporting this, the PostgreSQL database was used to store all paper metadata and to query results during analysis. In addition, GeoServer was used to visualize the map displays. Results were generated in the form of HTML, so can be visualized using any browser. Lastly, some other tools such as Microsoft Excel were used to generate, for example, the reference terms lists and to undertake some specific analyses. JavaScript graphing packages such as Chart.js and cytoscape.js libraries were used to provide specific graphical visualizations and some bespoke JavaScript was developed. The software used was predominantly open source.

In terms of computing, two Windows machines with Intel I9 processors, 64GB of memory and 2TB RAID SSDs as well as HDD backup storage were used as the main computing resource (with the database cloned to each machine).

Comparison of Effectiveness of the Results

As indicated a secondary goal of the experiment was to compare/correlate the results with the 'perceived technology status' for each year. Thus the above includes a measure of this.

To compare/correlate the technologies from the reference list with the outputs of the analysis, the occurrence profile of a term over time in research was superimposed with the occurrence year of the technology term from the reference. In addition, an absolute measure was produced of how many years before or after a term occurred in the reference list did it occur in published papers (see Section 3).

## 4. Results

The following presents the various measures, and the form of the results. The detailed output for each technology is available in HTML form alongside this paper.

The first and most basic measure (e.g., SOA, visualized in Figure 9) was the number of papers containing the technology term in each year—blue bars (the number was clipped at 100 to allow the initial point of growth to be seen more clearly). The actual occurrence numbers are available in a tabular output alongside the graph. The date the term first appeared in the reference was also included, for comparison purposes—red bar. The goal of this analysis was to provide a metric of the level of research applied to a term.

The next level of complexity was to assess in what subject areas the term was appearing. For example, was it occurring purely in technology-related papers (implying it was still in development), or was it also appearing in medical, social science or law papers, which might indicate progression into actual use? Because the subject area of the paper was not available in the metadata, a technique to try and identify it was developed. This involved creating 100 'keywords' for each subject area (e.g., for Medicine this might include 'operation', 'pathology'; for Law it might include 'case' or 'jury'). Depending on the score of these words, the paper could be ranked as say 60% law related, 20% technology. Both a

table and a graph were then produced for each technology showing this breakdown over time (Figure 10).

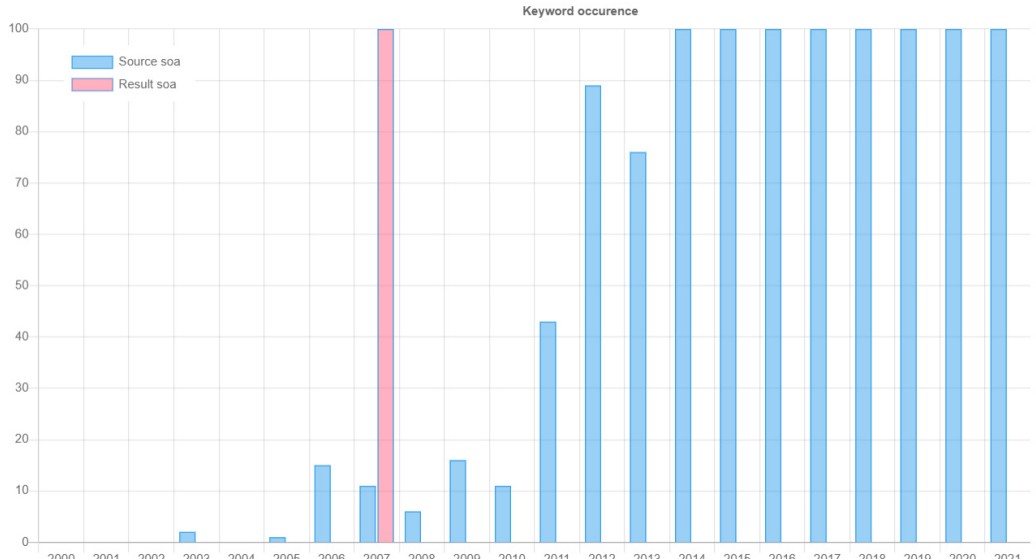

**Figure 9.** Occurrence SOA in paper abstracts (blue) and in the reference (in red). Note the scale is clipped at 100 to ensure the initial occurrences are identified.

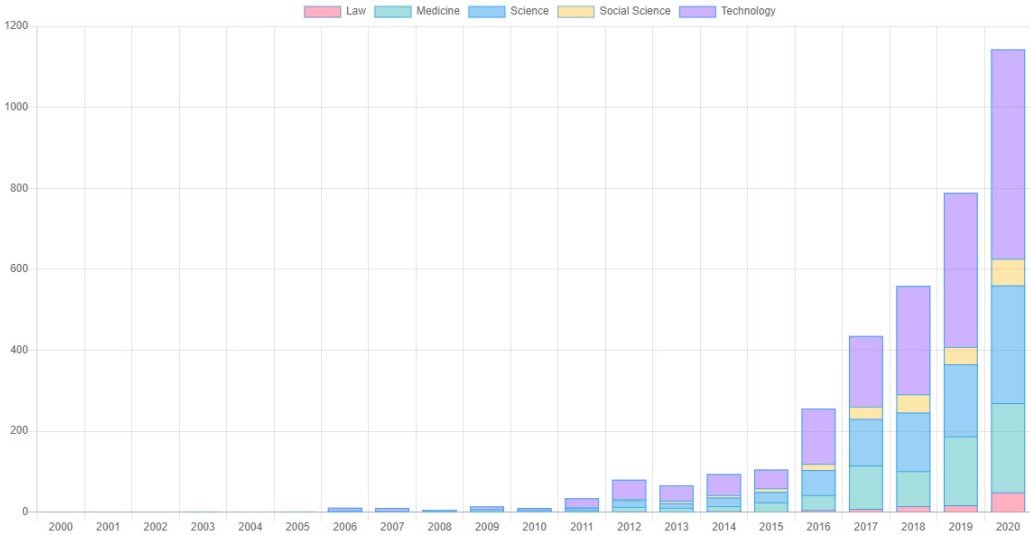

**Figure 10.** Breakdown of occurrences by domain (progression to actual use).

The paper abstracts containing the technology term were also categorized by country. The goal was to examine whether technology progression formed a particular pattern. Arthur (2009) [5] suggests that technology often forms in geographic pockets, meaning specific areas would show a high prevalence initially. Martin (2015) [37] has suggested this geographic focus only occurs for specific technologies, such as where physical resources are important (e.g., in drug development where specialist laboratory facilities are needed).

The country association to papers using the publisher library location was used as described in Section 3. This is therefore an approximation but does give an indication of where the research was undertaken (Figure 11).

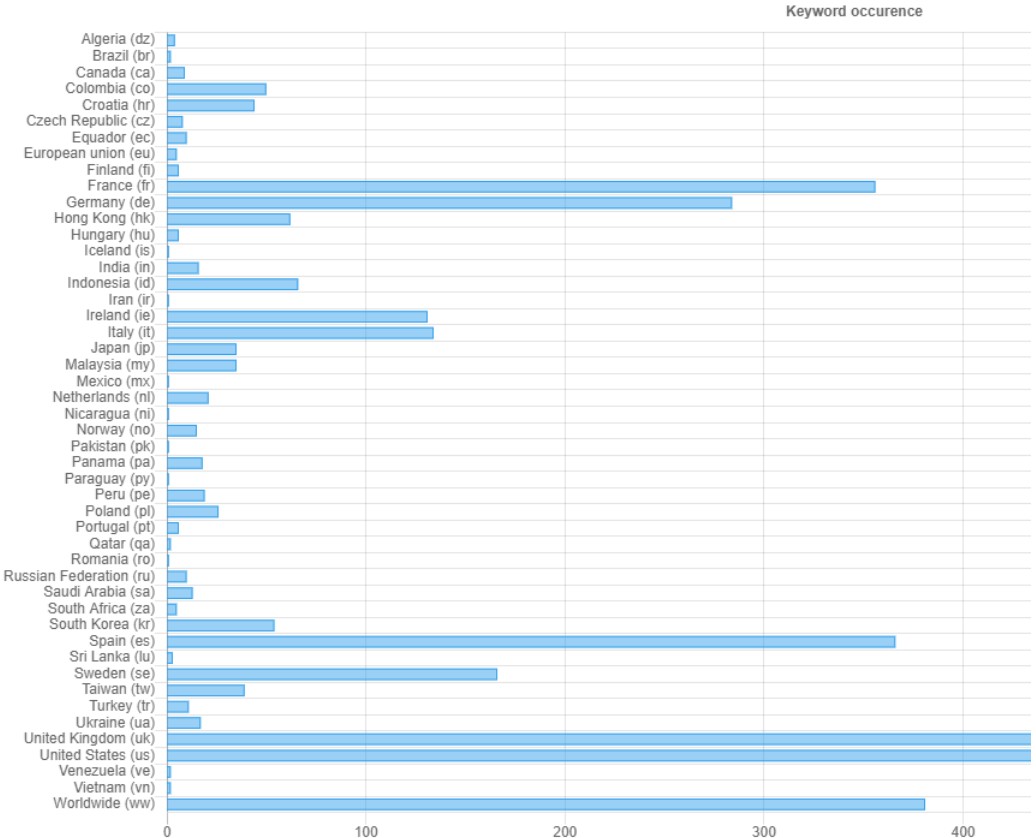

**Figure 11.** Occurrence of SOA in paper abstracts in different countries (Horizontal scale shows number of papers related to a topic per country).

The result was also visualized geographically (Figure 12); color was used to indicate geographic progression over time. The results show that some technologies have a dominant location from where they grow (an origin). In some cases, technologies remain tightly grouped geographically, but may start to spread to other locations because of the availability of specific researchers and skills in those geographic areas. This pattern is common for technologies which require a complex infrastructure—for example vaccine development and testing. Others spread quickly and uniformly after initial occurrence in a single location. Work by Schmidt (2015) [31] suggests just such an effect is likely to occur, suggesting that there is an element of 'mobility' in some technologies, for example IT related, compared to technologies which require significant research or production capability.

In Brackin et al. (2019) [16], the issue of technology groupings was a key element of technology progression. The example given specifically was the smartphone, and the relationship between technologies that form a cluster and progress in parallel. Looking for such clusters was identified as a useful measure. A technique was devised to calculate and visualize this. For each technology in the set being analyzed, the papers which contain the technology term searched for are known. Given those results, for each technology, it is possible to then see how many times each other technology in the set occurs in each technology set results. This amounts to an occurrence cross-product for each technology pair. If two technologies share no common papers, then no link is assumed. If two technologies share all, or a high number, a strong link is assumed. Figure 13 shows the result for SOA. The ***** symbol in column 2 indicates that this was the reference technology (original source of the papers) with which were searched for comparison with other technologies.

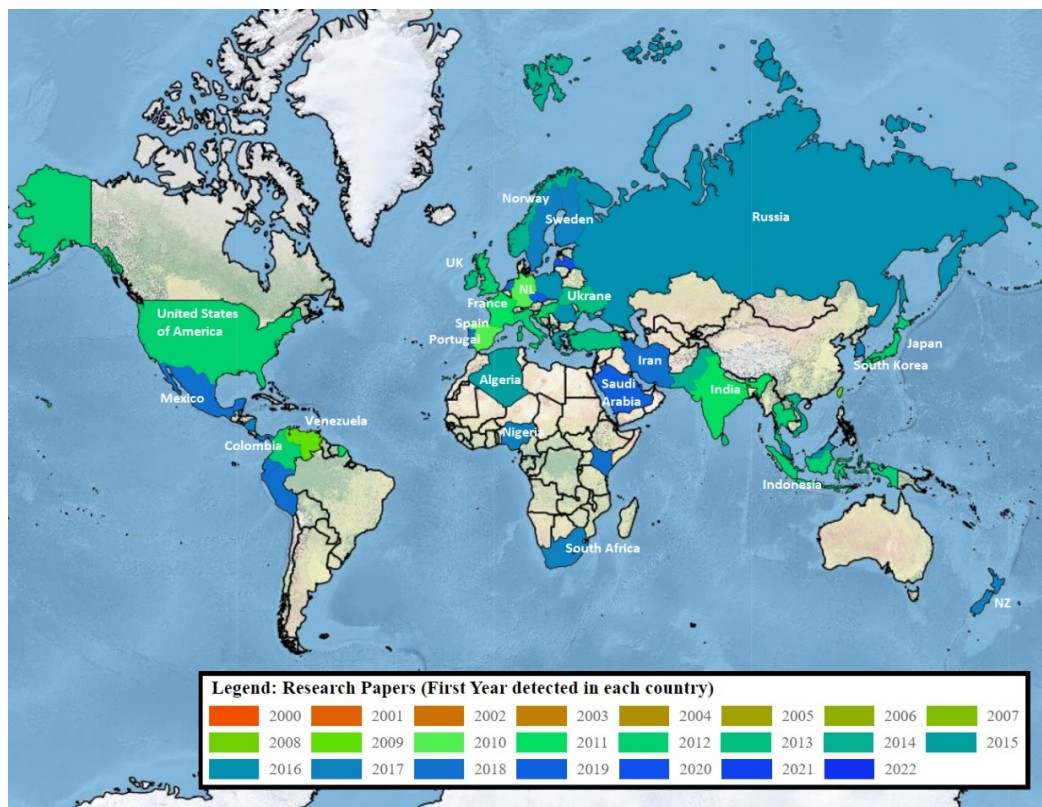

**Figure 12.** First occurrence of SOA by country, shown on a map backdrop.

| Other Term | Original | Descriptions it is present in |
|---|---|---|
| P2P | | 14 |
| Portlets | | 1 |
| Public Key Infrastructure | | 2 |
| Semantic Web | | 55 |
| Service Oriented architecture | | 315 |
| Smart Cards | | 2 |
| Smartphone | | 7 |
| SOA | ***** | 4854 |
| Syndication | | 1 |
| Text Mining | | 6 |
| Virtual Reality | | 4 |
| Virtual Worlds | | 3 |
| VoIP | | 4 |
| Wearables | | 2 |
| Web 2.0 | | 28 |
| Wikis | | 2 |
| XML | | 70 |

**Figure 13.** Detections of other topics within the SOA topic abstracts.

This result was also visualized graphically as a network graph—both for a specific technology topic, and as relationships between all technology topics. The technology link is shown by the presence of a line between the nodes (which are the technology topics), and the line thickness shows the level of commonality. Figure 14 shows the form of the diagram.

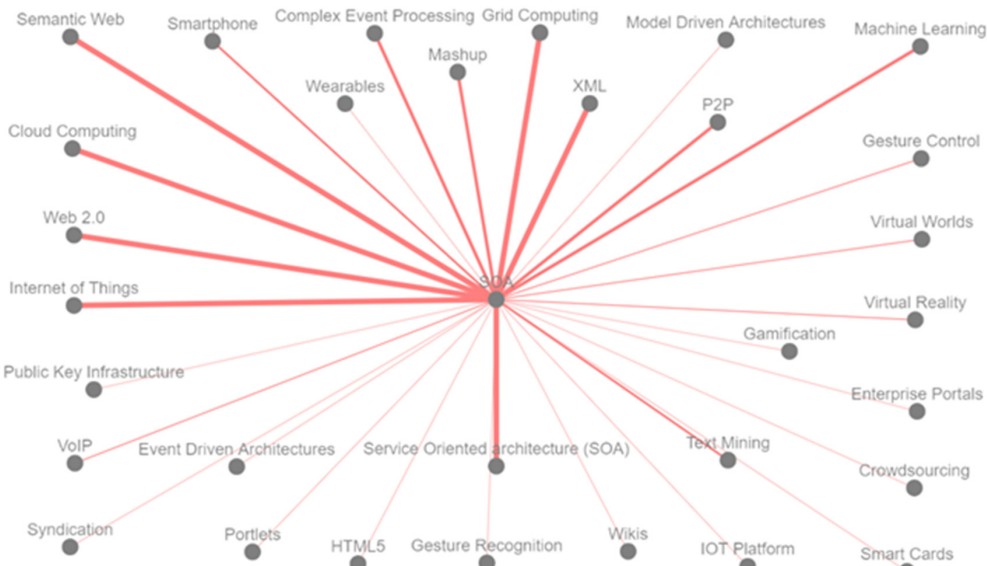

**Figure 14.** Graphical representation of relationships.

Only primary relationships are shown (i.e., technology topics directly related to the technology topic of interest). If the second level is added (topics related to that first set of topics), the diagram becomes significantly more complex.

The graph generation technology used does allow interactive browsing and so exploration of the full network is possible (as opposed to the simple hierarchical view shown in Figure 14).

Further composite analysis is potentially possible—for example looking at the geographic coincidence of technologies over time as well as their occurrence. Alternatively, one could analyze whether technology growth in a subject area occurs in a geographic pattern. Each of these analyses could offer further insight into technology progression. Additional measures are envisaged, which build on the comparison of technologies to demonstrate the principle of path dependence explored by Arthur (1994) [22], thereby allowing technology competition to be identified (relative growth).

In summary, a significant metadata resource was assembled for academic papers, and several analytical metrics were produced which corresponded to the common questions in relation to technology progression:

- How fast is a technology developing over time (showing interest in the technology)?
- In what fields is it gaining the most traction (showing the level of relevance in those fields)?
- How is a technology spreading geographically and are there particular geographic clusters?
- How is a technology linked to other technologies to form clusters?
- The metrics described here try to provide insight into each of these questions for a set of input technologies.

Taking topic areas, the technique of classifying data based on these does show value. Virtual and augmented reality use in medicine can be seen as the technology evolves. The results also show several papers emerging related to law and examining these they are clearly valid detections, with the papers addressing some of the legal and social aspects of augmented reality (e.g., public right infringement).

The geographic spread of technologies also shows value, although the lack of accurate origin of papers makes it less valuable, and it seems that the spread of technology topics is not particularly geographically bounded (although the analysis does allow some degree of pinpointing of the origin).

Lastly, the ability to visualize the relationships between technologies does allow an understanding to be drawn of clustering. It is quite clear that terms related to the smartphone such as internet of things, SOA, Web 2.0 and HTML5 are related technologies, as are technology clusters such as virtual reality, augmented reality, virtual worlds, and the smartphone are also strongly shown in this group too. This 'links' view also allowed synonyms/acronyms to be identified (e.g., Service Oriented Architecture and SOA are shown as strongly related on the graph).

The ability to drill down Into th" res'lts and see the actual detected technology terms in the abstracts, and to review the abstracts in a specific technology topic, was also useful. It allows the reviewer of the summary information to investigate specific results and identify false term detections and associations or points of particular interest.

The above results demonstrate that objective measures of technology progression, measured on various dimensions (time, space, application domain, co-occurrence with other technology growth) is possible using automated techniques. This confirms research question 2.

The framework does output a comparison of the input technologies and the profile of those technologies over time and space. This does allow an assessment of the technology and therefore an assessment of the reference.

The result of the analysis phase is 4 measures, 6 analyses most with both tabular and graphical representations/presentations for each of the 337 reference technology terms used as input. The analysis results page generated for each of these then provides links to 3370 artefacts (tables and graphs) generated automatically by the analysis process and accessible from the summary web page of the analysis, as shown in Figure 3.

For the reference technology terms list, the percentages of reference technology terms that could be detected in the academic paper abstracts was calculated, together with the difference between the point the term occurred in academic material, and the time it was seen in the reference technology terms list. Lastly, there is the total number of occurrences of a technology as a measure. Overall statistics were calculated; for the 337 terms, 75% were detected in academic abstracts. The detection failures seem largely to be in three-word technology terms and where there are potentially more likely names, for example: 'defending delivery drones', where 'drone defense' is a more likely generic technology term. Manual entry of alternative terms was included. a future iteration could potentially try to identify alternative name combinations (for example automated inclusion of acronyms (e.g., simply taking the first letter of each tech term, virtual reality and VR for example. This would not though associate Unmanned Aerial Vehicle (UAV) and drone. There were also potentially some technologies which were not matched because they simply did not progress through an academic route, for example Tablet PC.

The above largely provides visualizations of the assessments. There are though specific metrics available in the results. The first is the year of first occurrence (when the term was first detected) and the year when the rate of change of occurrence in papers was significant (in fact the threshold used for 'significant' was an increase of 100 paper detections per year).

There is also the time difference in years between a term occurring in the academic material (publication date), and the date in the reference technology terms list, was calculated. Additionally, whether the time was positive (academic detection occurred first), zero (detection was at the same time) or negative (the term occurred first in the reference list) was used to set one of three criteria–'precursor', 'the same time' and 'not precursor', respectively.

- 53% of technologies were discovered in academic paper abstracts prior to the occurrence date in the reference technology terms list.
- 8% were detected in academic papers at the same time as first seen in the selected media.

So overall and accepting that the input list was not created by the experiment and so its utility is limited at present, the automation does provide indicators in the same time range as occurrence in other sources. The analyses of geographic spread, subject area spread and relationships between technologies potentially offers additional insight.

The results also reinforce the idea that academic papers alone are not a sufficient measure of technology origin; an analysis capability requires multiple measures. Items such as Tablet PC were not detected before the reference, probably because its origins were industrial/commercial, rather than academic. This is highlighted by the fact that the tablet PC entered the Gartner model at a late stage too (i.e., it was not detected early by Gartner either). Similarly, with Bluetooth. Conversely, virtual reality, augmented reality and 3D scanners were all detected in papers considerably before noted on Gartner or in the Economist newspaper. These are composite technologies rather than individual innovations. The processing undertaken does also provide a considerable amount of detailed analysis of relationships not explicitly available from the reference sources.

Other quantitative measures were created in the output analytics. These are shown in Table 2. They provide an overview with the option of the user to drill down to examine the detail (as shown in the various visual representations in this section).

**Table 2.** Quantitative metrics of technology progression.

| Metric | Description of Metric | Unit of Measure |
|:---:|:---:|:---:|
| 1 | First Year of Detection of a technology in research papers. | Date |
| 2 | Year where significant detections occurred. | Date |
| 3 | Time difference between 1 and 2. | Interval |
| 4 | Total number of papers in which technology occurs | Count |
| 5 | Total number of countries in which technology research is identified | Count |
| 6 | Total number of domains/sectors (medicine, education etc) the technology term is identified with in research papers | Count |
| 7 | Number of strongly related technologies (based on a threshold of co-occurrences of 10) | Count |

In terms of the overall research question "To what extent is it possible to identify objective measures/indicators of technology progression using historic data in academic research", this question is addressed by the results above, i.e., the measures, although limited to academic research, do provide indicators of various aspects of technology progression in academic research. The results do also show several measures, for example the total number of occurrences of a technology topic as an absolute measure, or the number of occurrences in a given country or again the year in which the research occurrences was greater than 100.

In terms of the subsidiary element of the question, does this correlate with other methods, then an indication of this is possible (with the described before, same time and after measure against the reference technologies derived from sources such as Gartner and the Economist). It could be applied more rigorously if run as an on-going assessment using a team of experts operating using the normal method of panel assessment.

The final aspect of the experiment is whether the approach can offer objective validation of the theoretical concepts described in Section 2. Of the theories, an initial assessment of this is shown in Table 1.

Arthur (2009) [5] offers a narrative description of the nature of technology and the eco-systems that surround it. This research has realized that as a database model and populated parts of that model relating to research activity. Ongoing work will extend this to populate other elements of the model, for example industrial use of technologies. At that point the model would provide the basis for a continually updated model of multiple technologies' progression. The conclusion of the work so far is that Arthur's concepts do offer value in modelling technology progression and potentially the eco-systems around it.

Christensen (1997) [9] proposed the concept of disruptive technology. The indicators developed do seem to support the proposition that growth of technology is non-linear. In

most cases the graph of occurrences of a technology term within research papers shows a tipping point with initial 1–10 occurrences and then a growth in the next year or two peaking at many thousands. Examples of this include Service Oriented Architecture, Internet of Things, and Virtual Reality. Further work in progress which contrasts research growth with commercialization and looks at the relative timelines offers further insight into this aspect.

Mazzucuto's work in papers relating to state funding of research, again cannot be fully proven by this work but the indicators show that many technologies identified by consultants or the press did occur previously in research. A good example of this is Internet of Things.

Lastly, the concept of the platform, described by Langley & Leyshon (2017) [10] is borne out by the significant linking that occurs between groups of technology (measured by the number of paper abstracts in which multiple technologies are mentioned). Further work to show the alignment of growth of these connected technologies would reinforce this and assessing their adoption or relative commercial success.

## 5. Discussion

The goal of this work was to identify measures can be created using automated means and to help identify the direction of technology travel at a macro level (the research question). This has been demonstrated. The measures and processes require further refinement but were only intended as proof of concept. The general approaches used were in line with big data principles, as outlined in Mayer-Schoenberger & Cuckier (2013) [38]. The analyses used here provide an analytical view of technology progression based on academic paper metadata, which aligns with the outputs of manual forecasting techniques.

The techniques documented in this paper do not, on their own, offer a way to identify candidate technologies, or even significantly improve on human approaches typically used to rank technologies. They do offer the opportunity to potentially support the human view and provide extra insight and analysis to those undertaking technology forecasting. The analysis in this paper is based on one measurement point (academic research); multiple measures would be required for a more universal technology progression monitoring and forecasting capability. Measures of financial success of a technology, for example, would add another measure to indicate progression and further work in this area is the subject of a paper in preparation.

There is a rich collection of analyses of different aspects of technology growth as described in Section 1. Other research that could provide a theoretical basis for indicators include Gladwell (2000) [39] looking at the 'tipping point'; Langley (2014) [9], Simon (2011) [17] and Srnicek (2017) [18], looking at the concept of platforms; and Lepore (2014) [40], looking at the more negative aspects of progression.

The authors envision a series of monitors based on open data in various areas of the ecosystem and a unified model which can support the equivalent modelling undertaken in environmental and economic modelling. This paper is a first step in suggesting how such indicators could be constructed and most importantly integrated. There is in this, the chance to exploit the plethora of related big data, machine learning approaches which exist. Others have looked at different aspects of measuring technology progression, for example Carbonell (2018) [34] and Calleja-Sanz (2020) [32] and Dellermann (2021) [33]. In general, there are many opportunities for the application of big data and machine learning in this field, particularly with the model shown in Figure 1 as the basis.

Several approaches could be considered in taking this work from purely monitoring of technologies to a predictive capability. The first is to undertake an analysis of unusual words used in papers related to a topic. For example, detecting that 'virtual worlds' has recently occurred as a new term in paper abstracts. This could be done across the entire paper abstract set, or in existing topic areas. An initial version of this was produced, but it had a high level of false terms, as the libraries have been growing quickly (with lots of

organizations putting papers online). However, the application of big data techniques and machine learning could make an approach like this viable.

The technique for classifying papers based on a domain dictionary match (legal words, medical words, etc.) could be refined to detect a broader set of subject areas. The refinement of the dictionaries could also exploit machine learning as there are, for example, many libraries which contain only legal or medical papers, allowing training datasets to be efficiently created. The result would be richer classification information.

The use of the author/contributor information present in the paper metadata would also allow for the tracing of technology progression (authors typically have subject area specializations)-both in space and time. This was considered, but it does potentially have identity infringement issues, so was avoided in this initial research. There are also issues with ambiguity for the identification of authors (as the reference is typically surname and initial).

The most important next step is a unified, fully machine processable model based on the various sub-models described in Brackin et al. (2019) [16] and refinement to the point where the models can be created and informed by the sort of automated approach documented in this paper. The authors intend to continue this work in that direction. There is a strong base of overall (macro) approaches from Arthur (2009) [5] and Christensen (1997) [9], as well as several conceptual models for parts of the technology ecosystem identified by Mazzucato (2011) [6] and others on which to base a unified model.

The creation of further indicators which exploit other sources (perhaps the internet more generally) will be needed to support a unified model. Some of the techniques described in this paper will be valuable in creating these further indicators. Some will require new or automated approaches.

Lastly, more specific analysis of the results of this work would be valuable. This would help prove the results are useful. Insight into the temporal, spatial and subject area-related progression of specific technologies (for example autonomous vehicles) is an area that the approach could be applied to. Strambach (2012) [41] offers some existing analysis of the geography of knowledge, which could be further developed in terms.

### 6. Conclusions

The motivation of this paper came from experience in conducting and participating in technology progression assessment activities; and in these activities the authors recognized the diverse communities considering the problem (Figure 3) and their lack of integration. The authors have no doubt that big data, machine learning and neural networks all could make a significant contribution to assessments and represent a key opportunity to offer better insight. This paper, it is hoped, helps to not only identify approaches in these areas are viable but also that they can and should be integrated with both other areas in Figure 3 to offer the most value to those interested in technology progression assessment and monitoring.

**Author Contributions:** Conceptualization, R.C.B., M.J.J. and J.G.M.; methodology, R.C.B., M.J.J., A.L. and S.J.; software, R.C.B.; validation, R.C.B., M.J.J., A.L. and S.J.; formal analysis, R.C.B.; investigation, R.C.B. and A.L.; resources, R.C.B. and S.J.; data curation, R.C.B.; writing—original draft preparation, R.C.B. and M.J.J.; writing—review and editing, R.C.B., M.J.J., A.L. and S.J.; visualization, R.C.B.; supervision, M.J.J., A.L. and S.J.; project administration, R.C.B. and S.J.; funding acquisition, unfunded. All authors have read and agreed to the published version of the manuscript.

**Funding:** This research received no external funding.

**Data Availability Statement:** Data used in this paper was obtained from the Open Archive Initiative (www.oai.org). This is an open source. Other input data was entered manually. Results data is available from the consulting author on request. roger.brackin@nottingham.ac.uk.

**Conflicts of Interest:** The authors declare no conflict of interest.

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
