# Peer review of "Generating Indicators of Disruptive Innovation Using Big Data"

_futureinternet, doi:10.3390/fi14110327_

Round 1

Reviewer 1 Report

The title of the article formulates the author's task of predicting disruptive Innovation in the world. The abstract states that the author's approach "is expected to help improve the accuracy of assessments of technology progression over all". However, the article does not provide accuracy metrics or the author's forecast, there is no comparison of these metrics for different  methods. The methodology of the article is poorly traced. There are a lot of questions regarding the content of its application to the survey method used. No review of existing approaches and methods for evaluating innovations is given, nor is their critical analysis given. The reference does not reflect the current state in the field.

Separate remarks and shortcomings:

1. The article formulates 3 questions that need to be answered. However, the historical cross-section is not analyzed - how innovations were analyzed before, what methods were used, what are the advantages and disadvantages of these methods, what new opportunities appear in connection with the intellectual analysis of big data and networks

2. Many questions about the methods used in the article. On page 6, the authors say that they use a questionnaire to answer their first research question. At the same time, it does not describe how the questionnaire is formed, how the assessment of internal consistency, the discriminativeness of the test is carried out. How is the selection of experts and evaluation of the results of their work carried out?

3. For the second research question, the authors propose some indicators. At the same time, it is not given how these author's indicators are consistent with existing indicators for assessing innovation, how they differ from them. In addition, the author's indicators are presented in a qualitative form. In this case, it is not clear which scales are used to evaluate them. How are these indicators integrated? How then on this basis to build a predictive model?

4. On page 14, in the Results section, the authors describe the groups of respondents, but do not show how they are sampled - on what grounds, for what purposes were these differentiating features chosen?

5. The bibliography is very weak. It contains quite old publications, there are practically no scientific articles in this list.

I believe that this version of the article contains significant methodological flaws and cannot be published in the journal.

Author Response

Comments and Suggestions for Authors

The title of the article formulates the author's task of predicting disruptive Innovation in the world. The abstract states that the author's approach "is expected to help improve the accuracy of assessments of technology progression over all". However, the article does not provide accuracy metrics or the author's forecast, there is no comparison of these metrics for different  methods.  

Response:

Thank you for taking the time to review the paper, and for the helpful comments. Discussing these comments among the authors we concluded that the first issue to deal with is the use of ‘prediction’ in the title. This leads to several points raised in your commentary. Our intention was not to offer any form of ‘prediction’ but instead assessment of progression so far for a technology. Also, while the ultimate aim of our research is to improve the overall accuracy of assessments, the research within this paper does not have that as a goal. We will revise the title and introductory material to make this clear. We also consider the work somewhat embryonic in establishing the viability of indicators/measures and hence no significant assessment of accuracy has been undertaken. We see this as the next step. We will make sure this is also clear in the revised paper. 

Three of the authors have had extensive experience in running technology assessment panels and this paper’s origin and Brackin (2019) came from a desire to improve the process and outcomes. We have yet to find useful support tools or methods hence the lack of comparators.. All work we have located is within very specific and limited fields. For example a search of Future Internet only identified a few limited cases of comparable experimentation in specific fields. We are happy to hear from you in relation to work you feel we have missed, but at present we can’t identify it. 

The methodology of the article is poorly traced. 

Response:

We accept your assessment and have addressed it by including a more  extensive literature review, and clarifying the antecedence of the research presented which we believe now sets the methodological approach taken more in context of previous work.

Also we have seen various approaches to address the gap (for example NRC 2010,2011) which have failed to progress. We will make this clear in the paper. Lastly Brackin et al (2019) does provide much of the history of this research and so we chose not to repeat it here. However we will include more of the background. 

There are a lot of questions regarding the content of its application to the survey method used. 

Response:

 We appreciate that the survey is causing a significant number of issues. It was added as an adjunct to cover the issue of research relevance. In practice it is simply adding confusion. We therefore decided to remove it, which also provides more space to cover other issues raised and a simpler, single research question and subsequent structure. We will consider publication as a separate paper related to the questionnaire once the issues you identify can be covered. We think its removal refocusses the paper on the core topic.

No review of existing approaches and methods for evaluating innovations is given, nor is their critical analysis given. 

Response:  

The authors have undertaken or participated in horizon scanning activities and have reviewed available approaches and we have strengthened the references to the literature to reflect this... It could be that because of origins in the social science area we are missing some significant work, but  the NRC (2009, 2010) reports represent the best assessment we have seen. There are projects looking at other aspects for example patents and stock values but none we have found addressing research progression. We will though make it clear that we are basing our approach on filling an area with seemingly a dearth of objective approaches. 

The reference does not reflect the current state in the field.

Response: 

We acknowledge the current weakness in the literature review and have strengthened it. The field(s) this paper sits in is an intersection of two approaches and the additional references hopefully cover both areas more adequately

Separate remarks and shortcomings:

  1. The article formulates 3 questions that need to be answered. However, the historical cross-section is not analyzed 

Response: 

The current approach in every example we have seen is largely panel assessments or consultants such as Gartner or Deloitte. From the point of view of automated assessments, there are a handful, all specifically targeted, including those from Future Internet (e.g. progression of BitCoin) These will now be included. 

- what are the advantages and disadvantages of these methods,

Response: 

We will identify the primary advantages/disadvantages of using the proposed method over the manual approaches alone. 

- what new opportunities appear in connection with the intellectual analysis of big data and networks

Response: 

This is an excellent suggestion and we will elaborate that within the results and conclusions section.

  1. Many questions about the methods used in the article. 

On page 6, the authors say that they use a questionnaire to answer their first research question. 

Response: 

As stated above, the questionnaire has been withdrawn as it is peripheral. 

At the same time, it does not describe how the questionnaire is formed, how the assessment of internal consistency, the discriminativeness of the test is carried out. 

How is the selection of experts and evaluation of the results of their work carried out?

Response: 

As stated above, the questionnaire has been withdrawn as it is peripheral. 

  1. For the second research question, the authors propose some indicators. 

At the same time, it is not given how these author's indicators are consistent with existing indicators for assessing innovation, how they differ from them. 

Response: 

This is primarily because in the author’s experience there are very few indicators of technology progression. The Hype curve (Gartner) purports to be an indication but is ill defined. Technology Readiness is a potential standard indicator but has not been applied on a macro scale as far as we can see. These are described in Brackin (2019) but we will ensure this is clearly stated here. 

In addition, the author's indicators are presented in a qualitative form. In this case, it is not clear which scales are used to evaluate them. How are these indicators integrated? 

Response: 

Again a useful observation. In fact the measure of before/same time/after is a value (a three state measure). Also, the total number of occurrences of a technology is a quantitative measure. But we agree these measures are explicitly identified as quantitative. These numerical measures will be highlighted in the results. 

How then on this basis to build a predictive model?

Response: 

As stated above, the intention was not to suggest that the existing analysis could support a predictive model, but we agree the abstract/title are misleading in this respect. It is possible that elements, such as the technology growth over time within academic papers could be a predictive indicator. This will be discussed in a further paper (comparing commercial success with success in research). Within this paper we will clarify this point. 

  1. On page 14, in the Results section, the authors describe the groups of respondents, but 

do not show how they are sampled - on what grounds, for what purposes were these differentiating features chosen?

Response: 

As stated the questionnaire will be withdrawn from the text as described above. 

  1. The bibliography is very weak. It contains quite old publications there are practically no scientific articles in this list.

Response: 

The literature review/bibliography has been significantly enhanced. We hope this addresses the issue noted here. There is still the issue that the bulk of  research activity has occurred within the business and social science arena. Hence the dearth of such information rather than quantitative scientific papers, but there is some additional material that has been added.  . .  

Response: 

The current approach in every example we have seen is largely panel assessments or consultants such as Gartner or Deloitte. From the point of view of automated assessments, there are a handful, all specifically targeted, including those from Future Internet (e.g. progression of BitCoin) These will now be included. 

- what are the advantages and disadvantages of these methods,

Response: 

We will identify the primary advantages/disadvantages of using the proposed method over the manual approaches alone. 

- what new opportunities appear in connection with the intellectual analysis of big data and networks

Response: 

This is an excellent suggestion and we will elaborate that within the results and conclusions section.

  1. Many questions about the methods used in the article. 

On page 6, the authors say that they use a questionnaire to answer their first research question. 

Response: 

As stated above, the questionnaire has been withdrawn as it is peripheral. 

At the same time, it does not describe how the questionnaire is formed, how the assessment of internal consistency, the discriminativeness of the test is carried out. 

How is the selection of experts and evaluation of the results of their work carried out?

Response: 

As stated above, the questionnaire has been withdrawn as it is peripheral. 

  1. For the second research question, the authors propose some indicators. 

At the same time, it is not given how these author's indicators are consistent with existing indicators for assessing innovation, how they differ from them. 

Response: 

This is primarily because in the author’s experience there are very few indicators of technology progression. The Hype curve (Gartner) purports to be an indication but is ill defined. Technology Readiness is a potential standard indicator but has not been applied on a macro scale as far as we can see. These are described in Brackin (2019) but we will ensure this is clearly stated here. 

In addition, the author's indicators are presented in a qualitative form. In this case, it is not clear which scales are used to evaluate them. How are these indicators integrated? 

Response: 

Again a useful observation. In fact the measure of before/same time/after is a value (a three state measure). Also, the total number of occurrences of a technology is a quantitative measure. But we agree these measures are explicitly identified as quantitative. These numerical measures will be highlighted in the results. 

How then on this basis to build a predictive model?

Response: 

As stated above, the intention was not to suggest that the existing analysis could support a predictive model, but we agree the abstract/title are misleading in this respect. It is possible that elements, such as the technology growth over time within academic papers could be a predictive indicator. This will be discussed in a further paper (comparing commercial success with success in research). Within this paper we will clarify this point. 

  1. On page 14, in the Results section, the authors describe the groups of respondents, but 

do not show how they are sampled - on what grounds, for what purposes were these differentiating features chosen?

Response: As stated the questionnaire will be withdrawn from the text as described above. 

  1. The bibliography is very weak. It contains quite old publications there are practically no scientific articles in this list.

Response: 

The literature review/bibliography has been significantly enhanced. We hope this addresses the issue noted here. There is still the issue that the bulk of  research activity has occurred within the business and social science arena. Hence the dearth of such information rather than quantitative scientific papers, but there is some additional material that has been added.  . .  

Response: 

The current approach in every example we have seen is largely panel assessments or consultants such as Gartner or Deloitte. From the point of view of automated assessments, there are a handful, all specifically targeted, including those from Future Internet (e.g. progression of BitCoin) These will now be included. 

- what are the advantages and disadvantages of these methods,

Response: 

We will identify the primary advantages/disadvantages of using the proposed method over the manual approaches alone. 

- what new opportunities appear in connection with the intellectual analysis of big data and networks

Response: 

This is an excellent suggestion and we will elaborate that within the results and conclusions section.

  1. Many questions about the methods used in the article. 

On page 6, the authors say that they use a questionnaire to answer their first research question. 

Response: As stated above, the questionnaire has been withdrawn as it is peripheral. 

At the same time, it does not describe how the questionnaire is formed, how the assessment of internal consistency, the discriminativeness of the test is carried out. 

How is the selection of experts and evaluation of the results of their work carried out?

Response: As stated above, the questionnaire has been withdrawn as it is peripheral. 

  1. For the second research question, the authors propose some indicators. 

At the same time, it is not given how these author's indicators are consistent with existing indicators for assessing innovation, how they differ from them. 

Response: This is primarily because in the author’s experience there are very few indicators of technology progression. The Hype curve (Gartner) purports to be an indication but is ill defined. Technology Readiness is a potential standard indicator but has not been applied on a macro scale as far as we can see. These are described in Brackin (2019) but we will ensure this is clearly stated here. 

In addition, the author's indicators are presented in a qualitative form. In this case, it is not clear which scales are used to evaluate them. How are these indicators integrated? 

Response: Again a useful observation. In fact the measure of before/same time/after is a value (a three state measure). Also, the total number of occurrences of a technology is a quantitative measure. But we agree these measures are explicitly identified as quantitative. These numerical measures will be highlighted in the results. 

How then on this basis to build a predictive model?

Response: As stated above, the intention was not to suggest that the existing analysis could support a predictive model, but we agree the abstract/title are misleading in this respect. It is possible that elements, such as the technology growth over time within academic papers could be a predictive indicator. This will be discussed in a further paper (comparing commercial success with success in research). Within this paper we will clarify this point. 

  1. On page 14, in the Results section, the authors describe the groups of respondents, but 

do not show how they are sampled - on what grounds, for what purposes were these differentiating features chosen?

Response: As stated the questionnaire will be withdrawn from the text as described above. 

  1. The bibliography is very weak. It contains quite old publications there are practically no scientific articles in this list.

Response: The literature review/bibliography has been significantly enhanced. We hope this addresses the issue noted here. There is still the issue that the bulk of  research activity has occurred within the business and social science arena. Hence the dearth of such information rather than quantitative scientific papers, but there is some additional material that has been added.  . .  

I believe that this version of the article contains significant methodological flaws and cannot be published in the journal. 

Response: We are sorry to hear that you see the paper as that flawed. We believe that the revision being prepared may address many of the issues you raise, and in particular:

  •  dealing with the confusion of the title, 
  • improvement of the literature review/Clearer identification of antecedents
  • the removal of the questionnaire and a refocus of research question. 
  • the additional detail relating to the experiment   

Reviewer 2 Report

The article describes a study on the potential use of the information available in open publication repositories to monitor the progression of different technologies and even detect disruptive technologies.

The work is based on a very interesting idea and is approached convincingly. However, it is missing from the manuscript:

- A section that deepens the analysis of the correlation between the appearances of the technologies in the metadata of the collected publications and the "predictions" made in Gartner and the Economist. The question should be answered: The evolution of appearances in metadata follows the same trend as the annual appearance of the corresponding technologies in Gartner and Economist.

- A more detailed analysis of the indicators identified as possible detectors of different disruptive technologies.

Author Response

Comments and Suggestions for Authors

The article describes a study on the potential use of the information available in open publication repositories to monitor the progression of different technologies and even detect disruptive technologies. The work is based on a very interesting idea and is approached convincingly.

Response: Noted. Thank you for undertaking this review. We appreciate your extremely useful comments on this paper. We have tried to address them all and will provide a revised paper shortly.

However, it is missing from the manuscript:

- A section that deepens the analysis of the correlation between the appearances of the technologies in the metadata of the collected publications and the "predictions" made in Gartner and the Economist. The question should be answered: 

Response: The authors note your point, and we realise the current text is misleading in respect of the use of the Economist and Gartner. We should have been made clear that Gartner and the Economist were contributing to a list of reference technologies per year which is a representation of the state of knowledge at the time. Their predictions were not being assessed, and publishing such content would probably not be within copyright or even fairness rules. The authors envisage further research which would use a parallel manual and automated assessment to provide an authoritative assessment. This paper is simply demonstrating the viability of an approach.  

The evolution of appearances in metadata follows the same trend as the annual appearance of the corresponding technologies in Gartner and Economist.

Response: A point well made. This will be stated as the reference list typically shows the earliest identification (from either Gartner or the Economist). 

- A more detailed analysis of the indicators identified as possible detectors of different disruptive technologies.

Response: This is again a useful comment. The issue was mainly article length. We have chosen to remove the questionnaire and related research question from the paper. It is not core and will give us more room to address the issue you raise. We will elaborate as suggested. 

Reviewer 3 Report

General comment:

The paper tackles an interesting research topic on processes based on a big data approach for generating disruptive innovation metrics. Although it is recognized merit to the research developed so far, in formal terms, there are several opportunities for developing further the quality of the presentation of the results. The literature review needs severe improvement, also including previous studies published in your target journal. It is also recommended a general revision of the formal structure, as well as of the existent and new sections proposed to better frame the research questions raised in the scope of the current investigation.

Specific comments:

In order to increase the global quality of the manuscript, a set of recommendations and suggestions are made available below:

1.      The subtitle or running head could be eliminated.

2.      The introductory item needs to be completely revised, presenting the importance of the research questions, aims, state of the art, research questions, current findings, and structure of the paper.

3.      The literature review needs to be reinforced and diversified.

4.      There is a need for restructuring the section on Methodological Design.

5.      A final section on concluding remarks, limitations, implications and future research needs to be integrated into the formal structure of the paper.

6.      The references need to be formatted according to the MDPI submission guidelines.

Author Response

Comments and Suggestions for Authors

General comment:

The paper tackles an interesting research topic on processes based on a big data approach for generating disruptive innovation metrics. 

Although it is recognized merit to the research developed so far, in formal terms, there are several opportunities for developing further the quality of the presentation of the results. 

Response: Noted. We appreciate the time taken to review the draft and provide such helpful comments and particularly the roadmap in the last comment.  We have done our best to address them below and in a revised draft that will be available shortly. 

The literature review needs severe improvement, also including previous studies published in your target journal. 

Response: This point is noted, There was some urgency to submit, and we understood the limitations here. There is a significant amount of further references which were created post submission and now will be included. A number of papers identified in Future Internet were included but if there are others that we have missed please provide an indication of these. 

It is also recommended a general revision of the formal structure, as well as of the existent and new sections proposed to better frame the research questions raised in the scope of the current investigation.

Response: The paper was to some degree diluted with the Research Questionnaire, which is really a separate question. This has been removed, the sections revised and hopefully a clearer method and result review included. 

Specific comments:

In order to increase the global quality of the manuscript, a set of recommendations and suggestions are made available below:

  1. The subtitle or running head could be eliminated.

Response:Noted. We agree completely that the headers were getting confusing. The removal of the questionnaire and reduction to one research question makes a significant improvement here, but we will also reduce/carefully review the need for subheadings. 

  1. The introductory item needs to be completely revised, presenting the importance of the research questions, aims, state of the art, research questions, current findings, and structure of the paper.

Response: We agree with your thoughts here. In particular we realised the term prediction crept into the title and abstract, and this is somewhat misleading. While it may be an ultimate goal the paper is primarily focussed on technology progression assessment. We will try to refocus the initial material to more clearly identify the origins and motivations and deal with this issue. We also will review its structure against the MDPI framework. 

  1. The literature review needs to be reinforced and diversified.

Response: Agreed - revised and resubmitted paper will reflect this.We agree and work as described above is in progress to do this. If you have any specific literature suggestions you think may be relevant we would be delighted to consider these. The origins of the work came from undertaking/organising technology progression events over 10 years and examining their weaknesses. Thus we are less aware of all the seams of research related to internet analysis. 

  1. There is a need for restructuring the section on Methodological Design.

Response: The section on methodological design has been modified to reflect this and related comments.Thank you for pointing this out. We believe removal of the questionnaire may help this, but we will also examine this in more detail. The paper was re-formatted in the Future Internet format in a very limited time and so this didn’t help. But we will review other papers in FI to ensure we are consistent with it.   

  1. A final section on concluding remarks, limitations, implications and future research needs to be integrated into the formal structure of the paper.

Response: As above, this is noted and we will use the MDPI, FI titles as presented.

  1. The references need to be formatted according to the MDPI submission guidelines.

Response: Again helpful, as the intention was to comply. The lead author Agreed,  we will double check the format details and ensure the revision is correctly referenced/formatted. 

Round 2

Reviewer 1 Report

In its present form, the article is very different from the original version. It has been greatly improved. The reviewer's comments have been taken into account.

However, it is recommended to highlight the Conclusion in a separate paragraph, reflecting the contribution of the authors and noting the significant new opportunities appearing in connection with the intellectual analysis of big data and networks.

Author Response

Thank you for your comments. We have, as you recommended, added a paragraph to the conclusion/discussion section along the lines you suggested in the latest draft. 

Reviewer 2 Report

The authors have resolved the main drawbacks of the previous version of the article. It is now of sufficient quality for publication.

Author Response

Thank you for your further review. As there are no specific further suggestions we have not made any further changes related to this review. 

Reviewer 3 Report

Congratulations on the revised version of the manuscript.

Author Response

Thank you for your further review of the paper and positive comments. 

As there were no specific issues we have not made further changes in relation to this review.